# Analysis of Friction Noise Mechanism in Lead Screw System of Autonomous Vehicle Seats and Dynamic Instability Prediction Based on Deep Neural Network

**DOI:** 10.3390/s23136169

**Published:** 2023-07-05

**Authors:** Jaehyeon Nam, Soul Kim, Dongshin Ko

**Affiliations:** AI & Mechanical System Center, Institute for Advanced Engineering, Youngin-si 17180, Republic of Korea; jaehyeon@iae.re.kr (J.N.); soulkim@iae.re.kr (S.K.)

**Keywords:** friction noise, mode-coupling mechanism, squeal sensitivity map, squeal instability estimation, deep neural network (DNN)

## Abstract

This study investigated the squeal mechanism induced by friction in a lead screw system. The dynamic instability in the friction noise model of the lead screw was derived through a complex eigenvalue analysis via a finite element model. A two degree of freedom model was described to analyze the closed solutions generated in the lead screw, and the friction noise sensitivity was examined. The analysis showed that the main source of friction noise in the lead screw was the bending mode pair, and friction-induced instability occurred when the ratio of the stiffness of the bending pair modes was 0.9–1. We also built an architecture to predict multiple outputs from a single model using deep neural networks and demonstrated that friction-induced instability can be predicted by deep neural networks. In particular, instability with nonlinearity was predicted very accurately by deep neural networks with a maximum absolute difference of about 0.035.

## 1. Introduction

With the rapid development of autonomous vehicles, car seats in Level 4 systems and above are designed with structural freedom to serve various entertainment purposes rather than solely driving. Car seats are equipped with control systems that prioritize long forward movements and rotations, thus offering greater free movement rather than simply facing forward. Consequently, seats in autonomous vehicles require motorized systems for forward and backward movements; such systems typically consist of a combination of gears and motor systems. The control system for the forward and backward movements allows seat motion by using the force generated when the lead screw or lead nut is rotated by the motor. However, the noise generated, owing to friction between the lead nut and lead screw during seat movement, can potentially result in customer complaints.

Research on friction-induced vibration has mainly focused on improving brake squeals, which is known to be caused by a key mechanism called mode coupling. Studies on brake squeal enabled numerous researchers to understand the mechanism of friction-induced vibration and laid the groundwork for its application to the practical problem of squeak noise in various fields other than the automotive braking system [1,2,3,4,5].

The fundamental mechanism of autonomous vehicles is the propulsion provided by electric motors. Due to the masking effect of engine noise, the sensitivity of passengers to small noises originating from mechanical parts has increased. A fundamental solution is necessary for friction-induced vibrations, as they mainly occur in the high-frequency band. Issues related to friction between the lead screw and lead nut can cause friction-induced vibrations and even lead to gear noise due to dynamic gear backlash and wear-induced backlash caused by vibration between the lead screw and lead nut. Therefore, understanding and clarifying the squeal mechanism caused by the relative motion between the lead screw and lead nut in autonomous vehicle seats is a crucial issue that must be investigated.

Research on friction noise problems related to lead screws was conducted by employing analytical methods using experimental and simplified models [6,7,8]. Through eigenvalue analysis and experimentation, Olofsson et al. [9] predicted that the bending mode of the lead screw is the major mode causing friction-induced vibration and that the position of the lead nut is dependent on the friction noise. Kang et al. [10] analyzed the friction noise by using an experimental lead screw apparatus and demonstrated through modal experiments that friction noise occurs owing to the mode-coupling instability of the torsional mode and axial mode. Nam et al. [11] analyzed the characteristics of friction noise by using a lead screw and lead nut movement device and analyzed the friction noise mechanism occurring in the lead screw. However, these models were researched from the perspective of the friction noise vibration mechanism and could not fully reflect all the characteristics of the friction noise occurring in real systems. In particular, the analysis of the mechanism only reflects physical characteristics such as elastic coefficient, contact stiffness, and friction force, making it very difficult to provide direct solutions to problems occurring in production and on the road. Friction noise depends on the eigenmodes of the system that reflect geometric phenomena. It is essential to also consider the geometric shape at the contact surface, as it determines the static equilibrium state based on the distribution of the friction force [12,13,14,15,16]. However, it is impossible to mathematically reflect this accurately, and predicting nonlinear instabilities caused by changes in shape is extremely challenging. Therefore, by attempting the prediction of squeal instability occurring in the system from a deep learning perspective, we examined the squeal prediction performance and analyzed the results.

Recent leaps in artificial intelligence have made it possible to estimate and predict nonlinear data. Nam et al. [17] classified chaos, the most complex phenomenon in dynamical systems, using a convolutional neural network (CNN). Revach et al. [18] proposed a new Kalman filter structure, KalmanNet, using deep learning to learn the Kalman gain using deep learning rather than the linear approximation method used in conventional scaled Kalman filters, and showed that it outperforms traditional extended Kalman filters in estimation problems with nonlinear behavioral models. To develop an alternative for multivariate, nonlinear, and overdispersed modeling, Han et al. [19] reviewed deep learning techniques based on a multilayer perceptron to supplement the limitations of modeling in the civil engineering and transportation fields that had previously relied on traditional statistical techniques. Choi [20] proposed the local general regression neural network (LGRNN), a network structure synthesized from the LRNN and GRNN, to enhance the dynamic characteristics of the neural network while maintaining the features of the LRNN. Its effectiveness as a model for the non-linear prediction of non-static signals was reviewed, and it was shown to be more effective than traditional multilayer dynamic neural networks. Wang et al. [21] proposed a method for noise reduction in low-dose CT through deep learning, which showed considerable improvements in quantitative metrics and computation speed. Park et al. [22] trained a CNN–LSTM hybrid model that combines CNN and LSTM (long short-term memory) by using aircraft simulation data to predict the state variables. Lee et al. [23] presented a CNN compression technique using the local binary clustering method for MPEG-NNR and applied an additional nonlinear quantization technique on a regional basis after compressing the CNN using the conventional uniform quantization method. Kim et al. [24] introduced a sideslip angle estimation scheme by combining DNNs and nonlinear Kalman filters. The scheme was verified by both simulation and experimental results. Tork et al. [25] performed longitudinal–lateral dynamics control in an autonomous vehicle system. An adaptive neural network capable of producing nonlinear and complex mappings was designed. Šabanovič et al. [26] developed a road-type classification solution to improve vehicle dynamics control via the anti-lock braking system by estimating the friction coefficient using video data and DNN algorithms. Other studies are being conducted on the prediction and classification of diverse nonlinear signals, and various models are continuously being developed.

### Problem Formulation

In this study, we applied the mode-coupling mechanism [27,28,29,30], which is generally used in the brake system, to the lead screw system of an autonomous vehicle seat and investigated the friction noise that could occur between the lead screw and lead nut. In particular, we aimed to provide a method to understand the vibration/noise issues in the vehicle seat design stage by clearly understanding the squeal mechanism occurring in the lead screw through a friction-coupling integrated 3D model and a simplified model. The analytical model was validated by the results of the modal analysis of the lead screw in the free boundary condition. Then, the friction noise prediction was performed by a complex eigenvalue analysis of the lead screw system. To analyze the mechanism of friction-induced vibration in a lead screw system, a two degree of freedom model was constructed. Then, the friction noise in the lead screw system was analyzed by the mode-coupling mechanism. Additionally, we approached friction instability, which was solely reliant on physical interpretation, from a deep learning perspective and predicted the dynamic instability through deep learning. Therefore, this paper understood the friction noise generated by the lead screw from a mechanism perspective and described various methods to predict the friction noise. Based on the results, we proposed a way to improve the operating noise of autonomous vehicle seats at the design stage.

The paper is structured as follows: in Section 2.1, the result of a single component modal analysis and an experiment are verified; in Section 2.2, Section 2.3 and Section 2.4, the configuration of the finite element (FE) model, method of the complex eigenvalue analysis, and results of the analysis are described; in Section 2.5, the mode-coupling mechanism of the lead screw is described; in Section 2.6, the squeal noise is predicted using deep learning; in Section 3, there is a discussion of the results, followed by the conclusion.

## 2. Modeling and Results 

### 2.1. Single Component Modal Analysis and Experiment

Since complex eigenvalue analysis is conducted based on modal analysis, we verified the dynamic characteristics of the system through a single component modal experiment and analysis to enhance the reliability of the finite element model. Particularly, as the bending mode pair of the lead screw is predicted to be the main cause of instability due to friction coupling, we analyzed the bending mode pair of the lead screw and corrected the physical properties. The modal experiment was analyzed under free-end conditions, and the experimental setup is shown in Figure 1. The response point, where large deformation occurs under free-end conditions, was at the end of the lead screw on the bracket, and a three-axis accelerometer was attached to conduct the experiment.

The experimental results showed the frequency responses for the y and z directions’ impact hammer experiments to analyze the frequency difference between the bending mode pairs, as illustrated in Figure 2. Only the clear modes, where the phase plot changed by 180°, were represented as the fundamental modes. The maximum frequency difference between the paired modes was 6 Hz in the third bending mode. This difference in frequency is predicted to occur when the symmetrical condition breaks down owing to the bracket at the end. Regarding the mode shapes, although the z axis bending mode is predicted to approximate a pure beam’s bending mode, the y axis bending mode is predicted to exhibit a bending mode combined with the torsional mode owing to the bracket at the end.

The materials of the lead screw and lead nut were assumed to be steel, and the stiffness matrix was corrected based on the experimental results. The first and second modes are the first bending mode pairs, the third and fourth modes are the second bending mode pairs, and the fifth and sixth modes are the third bending mode pairs of the lead screw. The frequency difference between the mode pairs among the higher-order modes was large, consistent with the experimental results. The results of up to the sixth mode in the experiment and analysis are described in Table 1. The error for the final corrected single-component properties was within 3.3% in the third mode, and a complex eigenvalue analysis was conducted based on these results.

### 2.2. Finite Element Model

Figure 3 depicts an analysis model to analyze the squeal instability occurring in the lead screw system. The analysis model comprised a lead screw and lead nut, and the screw thread was modeled only for the length that overlaps with the lead nut for analytical efficiency based on the modal experimental results. The seat was assumed to move forward. The movement of the seat over time was simulated as the movement distance of the lead nut, and a complex eigenvalue analysis was conducted. Information on the modeled analysis parameters is presented in Table 2. If we assume that there is no friction curve, the rotational speed does not contribute to the instability of the system; hence, it was assumed to be 3 rad/s.

Assuming a frictional force due to the relative motion between the lead screw and lead nut, a normal force is orthogonally applied to the helix angle of the screw by the mass acting on the seat [31]. The external force applied to the lead nut is the same as the load applied to the two rails from the center of gravity of the seat weight. However, because the load from the steady-sliding equilibrium is determined as a term involving contact stiffness and becomes a homogeneous equation, a non-prestress analysis that reflects the load in stiffness was conducted. The contact force reflecting the contact stiffness and pseudo-rotation represents the static equilibrium state in linear stability analysis, and the analysis results are shown in Figure 4.

### 2.3. Method for Complex Eigenvalue Analysis of Finite Element Model

Various models such as the stick-slip model, sprag model, and mode-coupling model [32,33] were proposed to explain the mechanism of vibration/noise caused by friction. However, this study investigated the squeal mechanism of the lead screw, caused by mode-coupling instability. The mode-coupling mechanism was described in relation to the squeal mechanism, and the motion equation considering the friction caused by the relative motion between the lead screw and nut is expressed as follows:(1)$$\left({\bar{\lambda }}^{2}\left[\bar{M}\right]+\bar{\lambda }\left[\bar{C}\right]+\left[\bar{K}\right])\left\{\bar{y}\right\}=\{\bar{0}\right\}$$
where $\left[\bar{M}\right]$ represents the mass matrix, $\left[\bar{C}\right]$ is the damping matrix, $\left[\bar{K}\right]$ is the stiffness matrix, $\left[\bar{K}\right]$ is the eigenvalue, and $\left\{\bar{y}\right\}$ is the eigenvector corresponding to the eigenvalue. The stiffness matrix $\left[\bar{K}\right]$ becomes an asymmetric matrix owing to the friction force of the lead screw and lead nut, and the solution of the characteristic equation can be expressed as a complex number solution. The positive real part of the complex number solution indicates vibration instability caused by friction, implying a mode that triggers friction noise. Since it is very difficult to directly calculate the eigenvalues of an asymmetric matrix, the eigenvalue problem for an asymmetric matrix is solved using the eigenvalues of a matrix without damping and friction as follows:(2)$$\left(\lambda_{\mathrm{sym}}^{2}\left[M\right]+\left[K\right])\left\{z\right\}=\{0\right\}$$
where $\lambda_{\mathrm{sym}}^{2}$ is the eigenvalue of the matrix without damping and friction, and $\left\{z\right\}$ is the eigenvector corresponding to the eigenvalue. To solve the eigenvalue problem where damping and friction exist, the matrix of motion equations with damping and friction can be projected into the subspace of the eigenvector $\left\{z\right\}$, and a new matrix can be defined.
(3)$$\left[M^{*}\right]={\left[\left\{z_{1}\right\},\;\left\{z_{2}\right\},\;\ldots ,\left\{z_{n}\right\}\right]}^{T}[\bar{M}][\left\{z_{1}\right\},\;\left\{z_{2}\right\},\;\ldots ,\left\{z_{n}\right\}]$$
(4)$$\left[C^{*}\right]={\left[\left\{z_{1}\right\},\;\left\{z_{2}\right\},\;\ldots ,\left\{z_{n}\right\}\right]}^{T}\left[\bar{C}\right]\left[\left\{z_{1}\right\},\;\left\{z_{2}\right\},\;\ldots ,\left\{z_{n}\right\}\right]$$
(5)$$\left[K^{*}\right]={\left[\left\{z_{1}\right\},\;\left\{z_{2}\right\},\;\ldots ,\left\{z_{n}\right\}\right]}^{T}[\bar{K}][\left\{z_{1}\right\},\;\left\{z_{2}\right\},\;\ldots ,\left\{z_{n}\right\}]$$

Using the newly defined matrix, the eigenvalue problem can be expressed as follows:(6)$$\left(\lambda_{asym}^{2}\left[M^{*}\right]+\lambda_{asym}\left[C^{*}\right]+\left[K^{*}\right])\left\{y^{*}\right\}=\{0\right\}$$
(7)$$\left\{y\right\}={\left[\left\{z_{1}\right\},\;\left\{z_{2}\right\},\;\ldots ,\left\{z_{n}\right\}\right]}^{T}\{y^{*}\}\;$$

The major modes causing instability due to friction can be predicted based on the positive real part of the derived complex eigenvalue and the corresponding eigenmode.

### 2.4. Results of Complex Eigenvalue Analysis of Finite Element Model

Figure 5 shows the results of the complex eigenvalues with respect to the lead nut position. The friction coefficient was set to 0.5. The frequency trajectories with respect to the lead nut position tend to appear symmetrically around the center position of the screw. However, differences in frequency occur at both ends owing to asymmetric constraints in the y and z directions, which interfere with mode coupling. The first mode exhibits instability due to mode coupling at all positions in Zone A up to the center position of the lead nut. However, in the second mode, mode coupling is lost at the fifth position, preventing instability, but it reappears at the sixth and seventh positions. The third mode shows a frequency difference between the paired modes depending on the lead nut position, and the sensitivity of mode coupling varies with the nut position.

In the absence of a friction curve, the dynamic instability in the system solely occurs because of the instability due to mode coupling. Therefore, single-mode instability due to the negative slope of the friction curve does not occur. Figure 6 illustrates the changes in the frequency trajectory and the real part according to changes in the friction coefficient for each paired mode at the first position of the lead nut. When the friction coefficient is 0, each paired mode exists as an independent mode. However, because there is almost no frequency difference between the paired modes, mode coupling occurs and mode-coupling instability appears, even when the friction coefficient is very low at 0.05. Furthermore, the positive real part, indicating instability, increases as the friction coefficient rises. The corresponding mode shapes are shown in Figure 7. The occurrence of mode-coupling instability depends on the lead nut position because of the bracket constraints mentioned earlier. The sensitivity to instability is determined by the frequency difference between each paired mode based on the nut position because the lead screw causes the eigenmodes to appear. Moreover, the relatively high dynamic instability arising in the lead screw suggests an increased sensitivity to mode-coupling instability because there is almost no frequency difference between the paired modes, implying that it occurs because of a very low friction coefficient. To analyze these results from a mechanistic perspective, the sensitivity to mode-coupling instability between the paired modes was analyzed using a minimal model.

### 2.5. Lead Screw Squeal Instability Mechanism

#### 2.5.1. Description of Minimal Model

To analyze the vibration mechanism that includes friction originating from the lead screw, a simplified orthogonal pair model is depicted in Figure 8. Two orthogonal springs are attached to the mass, m, and two orthogonal contact springs are used at the contact point. Here, the two orthogonal springs simulate the bending mode pairs of the lead screw. Two vertical loads are applied, and the lead nut, simulated by the belt, rotates at a constant speed, V, acting on the contact surface. The motion equations for the two degrees of freedom for friction coupling can be written as follows:
(8)$$m\ddot{x}+\left(k_{1}+k_{c}\right)x={-\mu }_{x}k_{c}y+N_{1}$$
(9)$$m\ddot{y}+\left(k_{2}+k_{c}\right)x=\mu_{y}k_{c}x-N_{2}$$

For dimensionless time, $\tau =t\sqrt{k_{1}/m}$ was set, and the equation of motion corresponding to dimensionless is as follows:(10)$$x^{\prime \prime }+\left(1+k_{c}\right)x={-\mu }_{x}k_{c}y+N_{0}$$
(11)$$y^{\prime \prime }+\left(\varepsilon +k_{c}\right)y=\mu_{y}k_{c}x-N_{0}$$

Here, the superscript dash represents the differentiation with respect to $\tau (\geq \tau_{0})$, and the remaining parameters are defined as follows:(12)$$k_{c}\equiv k_{c}/k_{1}$$
(13)$$\varepsilon \equiv k_{2}/k_{1}$$
(14)$$N_{1}=N_{2}=N,\;N_{0}\equiv N/k_{1}$$

In the steady-sliding equilibrium ($x^{\prime \prime }=y^{\prime \prime }=x^{\prime }=y^{\prime }=0$), a solution was assumed to describe the motion equation as follows. If the friction force is defined as a function of speed, the friction curve can be expressed at the two contact points of the mass in the steady-sliding equilibrium.
(15)$$x=u+u_{eq},\;y=v+v_{eq}$$
(16)$$\mu_{x}=sgn(V-u^{\prime })(1-e^{d\left|V-u^{\prime }\right|})(\mu_{k}+\left(\mu_{s}-\mu_{k}\right)e^{-h\left|V-u^{\prime }\right|})$$
(17)$$\mu_{y}=sgn(V-u^{\prime })(1-e^{d\left|V-v^{\prime }\right|})(\mu_{k}+\left(\mu_{s}-\mu_{k}\right)e^{-h\left|V-v^{\prime }\right|})$$
(18)$$\mu_{0}={\left.\mu_{x}\right|}_{u^{\prime }=0}={\left.\mu_{y}\right|}_{v^{\prime }=0}$$

Here, the equilibrium point ($u_{eq},\;v_{eq}$) of the two-DOF friction-coupled model is given as follows:(19)$$u_{eq}=N_{0}(\varepsilon +k_{c}+k_{c}\mu_{0})/(\varepsilon +k_{c}+\varepsilon k_{c}+k_{c}^{2}+k_{c}^{2}\mu_{0}^{2})$$
(20)$$v_{eq}=-N_{0}(k_{c}-k_{c}\mu_{0}+1)/(\varepsilon +k_{c}+\varepsilon k_{c}+k_{c}^{2}+k_{c}^{2}\mu_{0}^{2})$$

To analyze the stability of the solution of the motion equation including the friction curve, a Taylor expansion near the equilibrium point of the nonlinear friction curve, which allows the representation of the linearized friction curve, can be used.
(21)$$\mu_{x}=\mu_{0}+{\left.\frac{{\partial \mu }_{x}}{{\partial u}^{\prime }}\right|}_{u^{\prime }=0}\cdot u^{\prime }+O_{2},\;{\left.\frac{{\partial \mu }_{x}}{{\partial u}^{\prime }}\right|}_{u^{\prime }=0}\cdot k_{c}\equiv \alpha$$
(22)$$\mu_{y}=\mu_{0}+{\left.\frac{{\partial \mu }_{y}}{{\partial v}^{\prime }}\right|}_{v^{\prime }=0}\cdot v^{\prime }+O_{2},\;{\left.\frac{{\partial \mu }_{y}}{{\partial v}^{\prime }}\right|}_{v^{\prime }=0}\cdot k_{c}\equiv \beta$$

The motion equation near the equilibrium point is as follows: (23)$$u^{\prime \prime }+\alpha u^{\prime }+\left(1+k_{c}\right)u+\mu_{0}k_{c}v=0$$
(24)$$v^{\prime \prime }+\beta v^{\prime }+\left(\varepsilon +k_{c}\right)v-\mu_{0}k_{c}u=0$$

Here, the solution can be derived as explained below; it was assumed that there are no slopes of the two linearized friction curves from the closed form eigensolution. Assuming that only instability due to friction coupling exists, the eigenvalues are expressed as follows from the characteristic equation.
(25)$$\left.\left\{\begin{array}{c} u\\ v \end{array}\right.\right\}=Ve^{\lambda t}$$
(26)$$\det\left(\lambda^{2}\left[I\right]+\left[K_{\mathrm{sym}}\right]+\mu_{0}k_{c}\left[K_{\mathrm{asym}}\right]\right)=0$$
(27)$$\left[K_{\mathrm{sym}}\right]=\left[\begin{array}{cc} \Omega_{u}^{2} & 0\\ 0 & \Omega_{v}^{2} \end{array}\right]$$
(28)$$\left[K_{\mathrm{asym}}\right]=\left[\begin{array}{cc} 0 & E_{uv}\\ E_{vu} & 0 \end{array}\right]$$
(29)$$\lambda_{uv}=a\pm \sqrt{b^{2}+\mu_{0}^{2}k_{c}^{2}E_{uv}E_{vu}}$$
(30)$$a={(\Omega }_{u}^{2}+\Omega_{v}^{2})/2$$
(31)$$b={(\Omega }_{u}^{2}-\Omega_{v}^{2})/2$$

Here, the eigenvalue $\lambda_{uv}$ is complex, and, if the real part is positive, it is referred to as mode-coupling instability. The necessary and sufficient conditions for dynamic instability are $\sqrt{b^{2}+\mu_{0}^{2}k_{c}^{2}E_{uv}E_{vu}}$, and the necessary condition is $E_{uv}E_{vu}<0$. However, because the necessary condition is satisfied in Equations (16) and (17), the system may exhibit dynamic instability, and when the stiffness ratio ($\varepsilon =(k_{2}/k_{2})$) is 1, the system meets the necessary and sufficient conditions. This implies that noise vibration due to friction can occur because the sensitivity to mode-coupling instability increases as the frequencies of the lead screw’s paired mode become closer.

#### 2.5.2. Analysis Results of Degree of Freedom of Minimal Model Mechanism

The aim of this study was to analyze the mechanism of frictional vibration occurring due to mode-coupling instability in the lead screw system. Figure 9 shows the results of the sensitivity analysis of the stiffness ratio of the minimal model for friction coupling. The frequency trajectory shows the results according to changes in the stiffness ratio when the friction coefficient is zero. The real part represents the positive real part generated by changes in the stiffness ratio and friction coefficient. We performed numerical analyses with the stiffness ratio parameter ranging from 0.8 to 1.2 and the friction coefficient ranging from 0 to 0.5. As the stiffness ratio increases, the frequency of one of the paired modes increases, and, when the stiffness ratio reaches 0.9, dynamic instability appears even at a low friction coefficient. The moment the stiffness ratio becomes 1 (point “A”), the frequencies of the paired modes become identical, and sensitivity to dynamic instability reaches its peak. If the stiffness ratio becomes larger than 1, the frequencies of the paired modes diverge again, with each mode having an independent frequency, and dynamic instability disappears at a stiffness ratio of 1.2. This implies that friction noise can occur if the stiffness ratio of the lead screw is between 0.9 and 1.1.

Figure 10 shows the mode-coupling mechanism with respect to the friction coefficient when the stiffness ratio is 0.95. Figure 10a,b show the frequency trajectory and real part according to changes in the friction coefficient, respectively. Two very adjacent independent paired modes combine into one mode as the friction coefficient increases and reaches the threshold friction coefficient (approximately 0.25), producing a positive real part. This is the onset time when friction noise occurs. This suggests that systems such as the lead screw, which have very close paired modes, can cause mode-coupling instability due to very low friction forces.

The stability map for the set mode parameters is shown in Figure 11. As seen in Figure 11a,b, the two frequencies of the paired modes cause instability owing to the friction coefficient when the stiffness ratio is very close. Especially when the stiffness ratio is around 1 (when the frequencies of the pair modes are the same), the size of the positive real part is at its maximum for all friction coefficients. Since systems designed around the rotational axis necessarily show paired modes, ensuring system stability through structural damping and radiation damping effects could be a design solution to reduce noise caused by friction.

### 2.6. Deep Neural Network Based Squeal Instability Estimation

#### 2.6.1. Data Augmentation

Theoretical studies on squeal instability may not fully reflect complex geometric shapes, even with sophisticated models [27]. In contrast, analyses using FE models can intricately reflect complex geometric shapes, but complex systems have issues related to analysis time and convergence, even with linear analysis. Therefore, performing a squeal analysis that reflects complex geometric parameters can be challenging. Furthermore, obtaining numerous results from repeated squeal-based experiments is impossible, making it challenging to construct a big data dataset.

Regression analysis based on the experimental design method is a highly advantageous approach when building a dataset, as it can derive a large number of results within a certain range with few iterations. However, the experimental design procedure is based on linearity, making it difficult to guarantee a significant *p* value when the squeal instability includes nonlinearity. This section seeks to provide a method for predicting even difficult results that include nonlinearity, such as squeal noise, based on deep learning. Therefore, we applied the spline technique as the interpolation method for data augmentation to reflect the nonlinearity as accurately as possible [34]. The 3D dataset augmented by the spline method is shown in Figure 12. Figure 12a–c refer to the 1st, 2nd and 3rd augmented bending modes, respectively.

The augmented squeal dataset has characteristics that include very high nonlinearity, such as the shape of the experimental model, instead of parameters that can predict the instability, such as the elastic coefficient, contact stiffness, friction coefficient, and friction curve slope. This aids in the simulation of the changes in the experimental conditions that include extreme nonlinearity, uncertainties, and environmental conditions. Since a large dataset is essential for improving machine learning accuracy, data augmentation is performed using various methods. Previous machine learning research showed that the results were not greatly influenced when there were over 1000 data points; hence, 2500 data points were derived. The composition of the dataset is shown in Table 3 [35].

#### 2.6.2. Squeal Noise Prediction Using Deep Learning

We constructed a model according to the data shown in Table 4 to predict the squeal instability with the DNN using the dataset augmented by nonlinear spline interpolation. The aim of this research was to demonstrate that deep learning can predict squeal analysis that considers complex factors such as design variables, provided the dataset is sufficient. Hence, we configured the DNN model with a simple architecture rather than a complex one.

Since the results we aimed to predict consist of three positive real numbers, we designed the DNN model to have multiple outputs using a single architecture. We used the ReLU function as the activation function and the gradient-descent-based Adam optimizer as the optimization function to achieve rapid convergence. The initial weights are critical, as they determine the convergence speed and improvement in accuracy. While there are various algorithms for setting the initial weights, the method that shows the best correspondence with the ReLU function is the He Gaussian initialization method [36]. However, this study did not use an initialization method because the data were relatively simply structured, and the architecture was simplified.

We identified the characteristics of the unstable real part of the design variables in the training data and simultaneously indicated the real time errors through the validation data. We then performed tests utilizing data not used in the training to obtain the final error results. We set the batch size for learning to 10 and the learning rate of the optimization function to 0.001. Continuous learning of sequential data reduces overfitting and loss of validation data. Therefore, we used a callback function to prematurely terminate the training and derive an optimized model if there were no improvements in the validation loss during 1000 training iterations. We also examined the reliability of the vibration instability analysis using deep learning through five sets of results randomly shuffled from the same data. Figure 13 shows the training process of the model and the results predicted through the trained model.

Figure 14a–c show the results for the first, second, and third paired modes, respectively. The prediction results showed very close approximations to the correct answers without significant errors across all modes. The average absolute differences of the predicted values for all results are presented in Figure 15. Although the fourth result showed a relatively large error, the maximum absolute error value was below 0.035. This suggested that the dynamic instability for geometric nonlinearity can be predicted to closely match the analyzed results using deep learning without relying on complex analysis methods.

Figure 10 shows the mode-coupling mechanism with respect to the friction coefficient when the stiffness ratio is 0.95. Figure 10a,b show the frequency trajectory and real part according to changes in the friction coefficient, respectively. Two very adjacent independent paired modes combine into one mode as the friction coefficient increased and reached the threshold friction coefficient (approximately 0.25), producing a positive real part. This was the onset time when friction noise occurs. This suggested that systems such as the lead screw, which have very close paired modes, can cause mode-coupling instability due to very low friction forces.

## 3. Discussion and Future Work

In this study, we constructed a squeal model by applying a vibration instability analysis algorithm, typically used to analyze brake squeal, to a lead screw analysis model and conducted a vibration instability analysis to examine the squeal mechanism occurring in lead screws. We also researched a methodology that could predict not only the nonlinear results but also multiple outputs through a single model by forecasting the instability of the lead screws owing to the geometric shape deformation from using a DNN. According to the results of a complex eigenvalue analysis of the lead screw, the system’s instability was manifested in the bending mode pairs. This is because the paired modes of the lead screw characteristically have almost no frequency difference; hence, the mode-coupling sensitivity was large even at a low friction coefficient. The analysis of the mode-coupling sensitivity showed that the friction noise generated by the lead screw occurs when the ratio of the two stiffnesses is between 0.9 and 1. The mode-coupling sensitivity was higher when the stiffness ratio is 1. This means that the natural frequencies of the paired modes were completely coincident, suggesting the need to separate the natural frequencies of the paired modes through constraints and geometry changes in order to stabilize the friction noise. We also demonstrated that the single architecture multi-output model using a DNN could rapidly predict the instability of models with extreme nonlinearity, such as geometric shape nonlinearity. The analysis showed that the absolute difference between the predicted values was very accurate, up to 0.035. This implies that even a simple DNN model can be used to conduct complex instability analyses of systems containing nonlinearity.

In future work, we intend to conduct studies applying machine learning to transient analysis that can predict the actual behavior of friction noise. There are two main ways to estimate friction-induced vibration. The first method can be predicted based on the onset time of dynamic instability using the linearization. The second method is calculated by a differential equation to consider all of its non-linear characteristics. A direct solution to the differential equation can yield a more accurate solution, but the nonlinearity makes the computation complex and time-consuming. Therefore, we plan to use machine learning to approximate the solution of the friction-induced vibration.

## Figures and Tables

**Figure 1 sensors-23-06169-f001:**
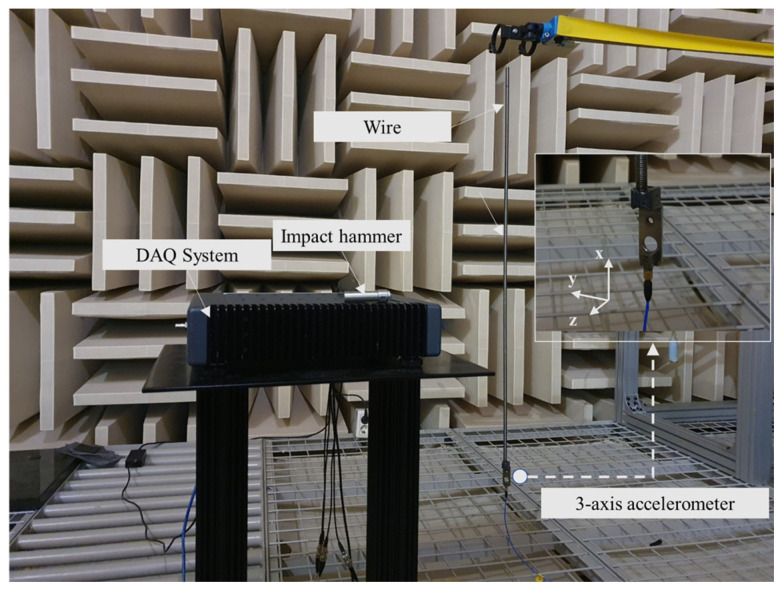
Test setup.

**Figure 2 sensors-23-06169-f002:**
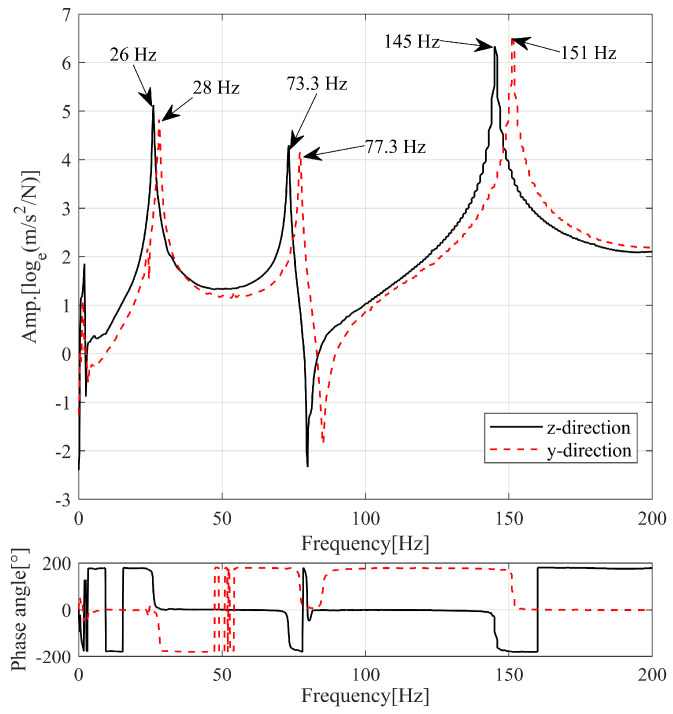
Results of the modal test.

**Figure 3 sensors-23-06169-f003:**
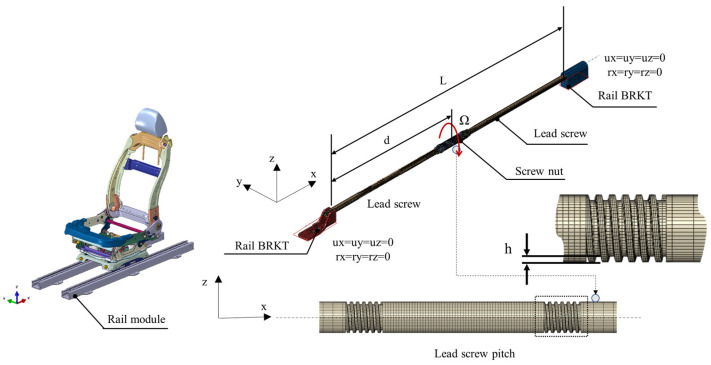
Configuration of FE Model.

**Figure 4 sensors-23-06169-f004:**
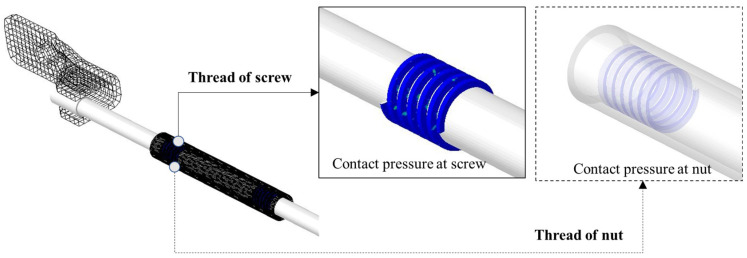
Contact force distribution on lead screw.

**Figure 5 sensors-23-06169-f005:**
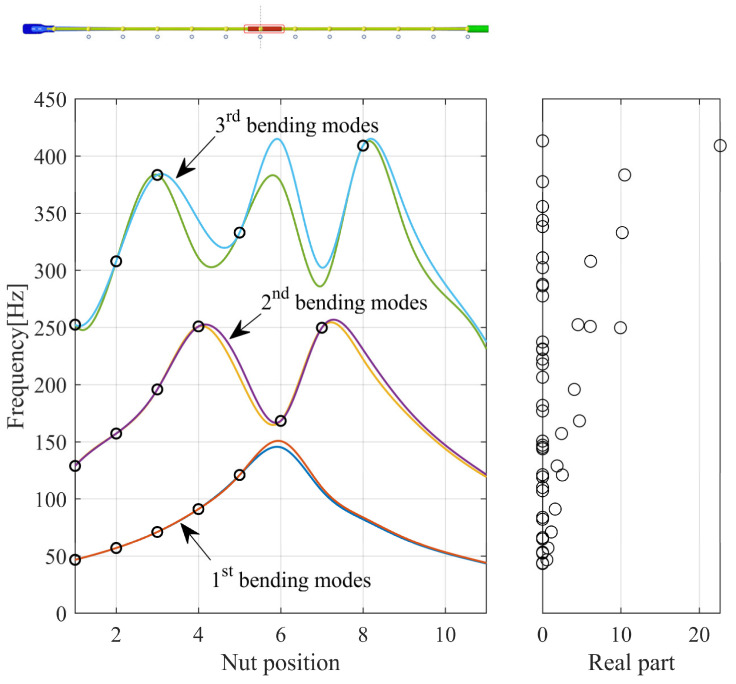
Frequency loci with lead nut location. The empty circles indicate the positive real parts.

**Figure 6 sensors-23-06169-f006:**
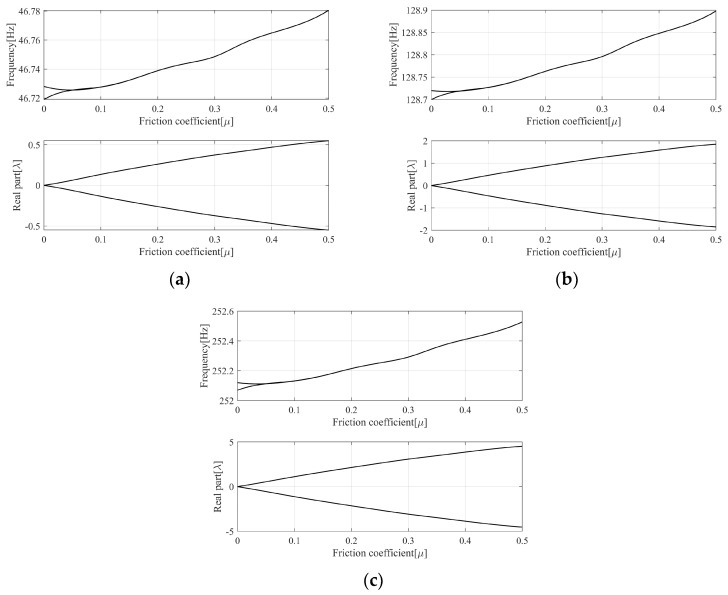
Loci of frequencies and real parts with respect to friction coefficient: (**a**) 1st bending modes; (**b**) 2nd bending modes; (**c**) 3rd bending modes.

**Figure 7 sensors-23-06169-f007:**
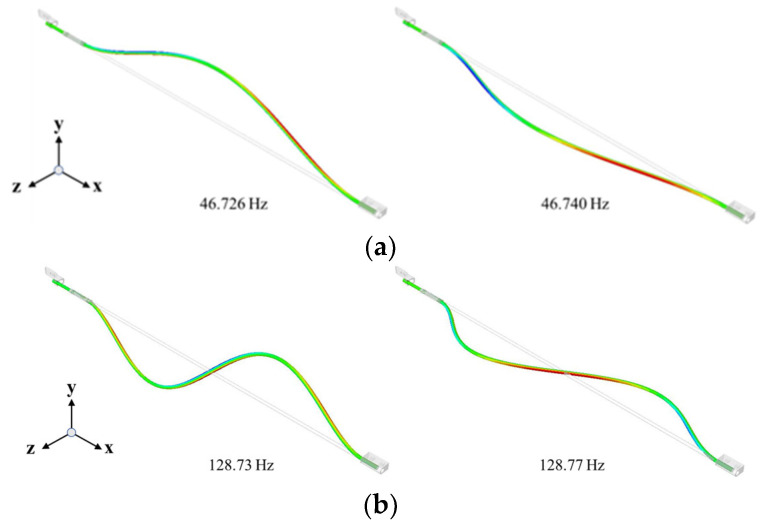
Mode shapes: (**a**) 1st bending modes; (**b**) 2nd bending modes; (**c**) 3rd bending modes.

**Figure 8 sensors-23-06169-f008:**
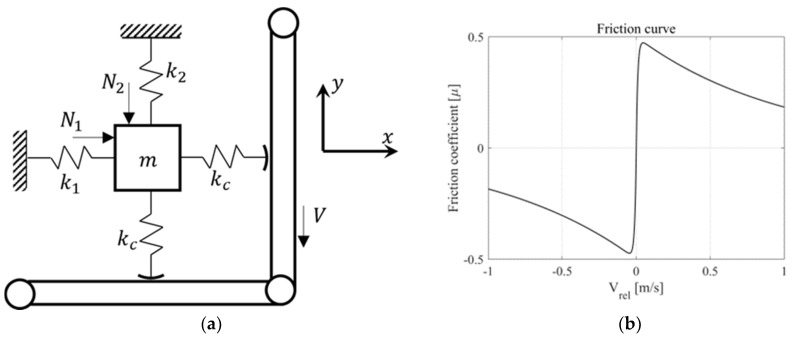
Description of friction-induced model: (**a**) 2 DOF minimal model; (**b**) friction curve.

**Figure 9 sensors-23-06169-f009:**
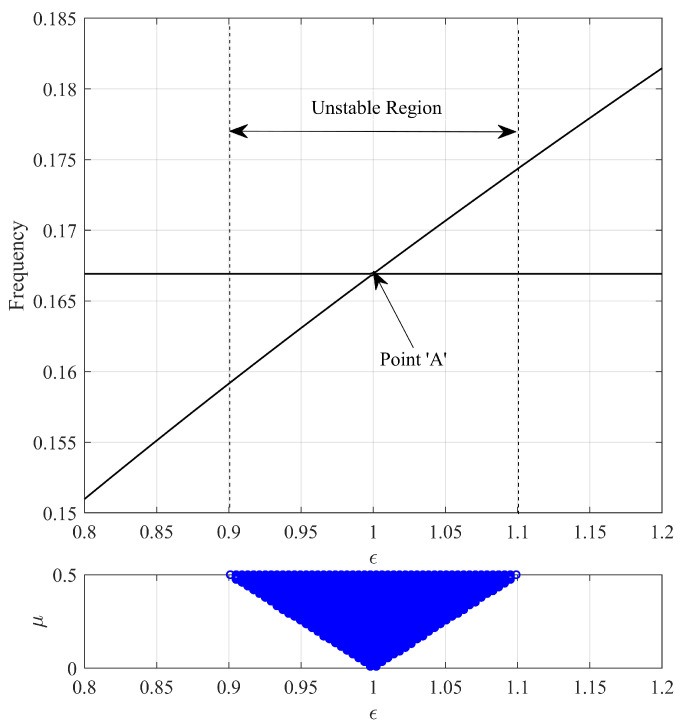
Eigenvalue sensitivity analysis; “o” denotes positive real part.

**Figure 10 sensors-23-06169-f010:**
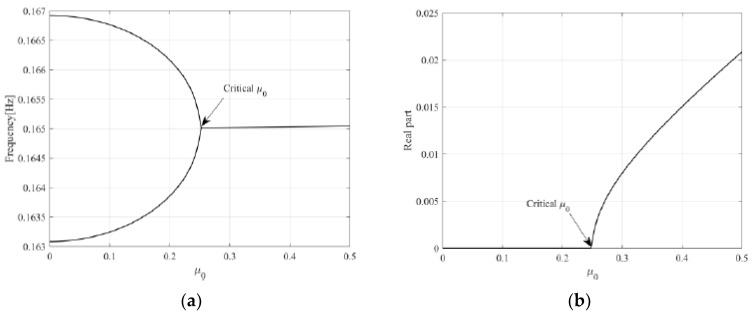
Mode merging with respect to friction coefficient for ε = 0.95: (**a**) frequency loci; (**b**) real part.

**Figure 11 sensors-23-06169-f011:**
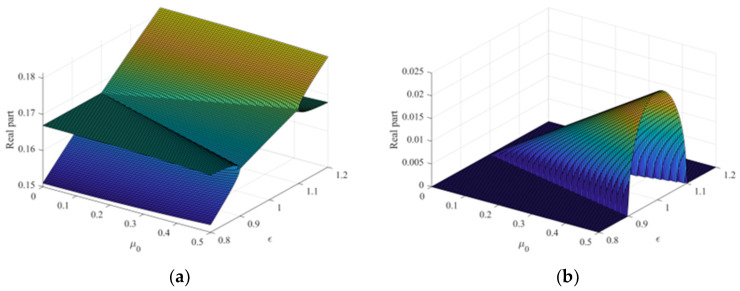
Stability map for the system parameter: (**a**) frequency loci; (**b**) real part.

**Figure 12 sensors-23-06169-f012:**
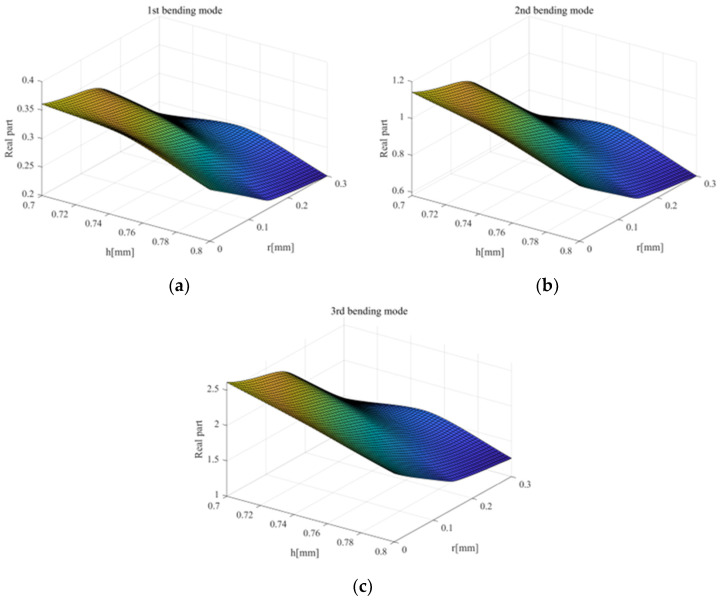
Dataset augmentation.

**Figure 13 sensors-23-06169-f013:**
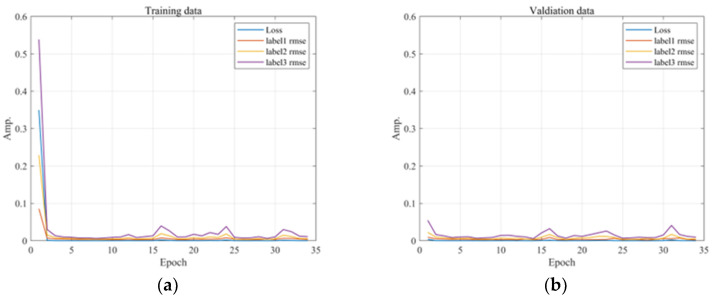
Performance comparison of unstable modes: (**a**) training dataset; (**b**) validation dataset.

**Figure 14 sensors-23-06169-f014:**
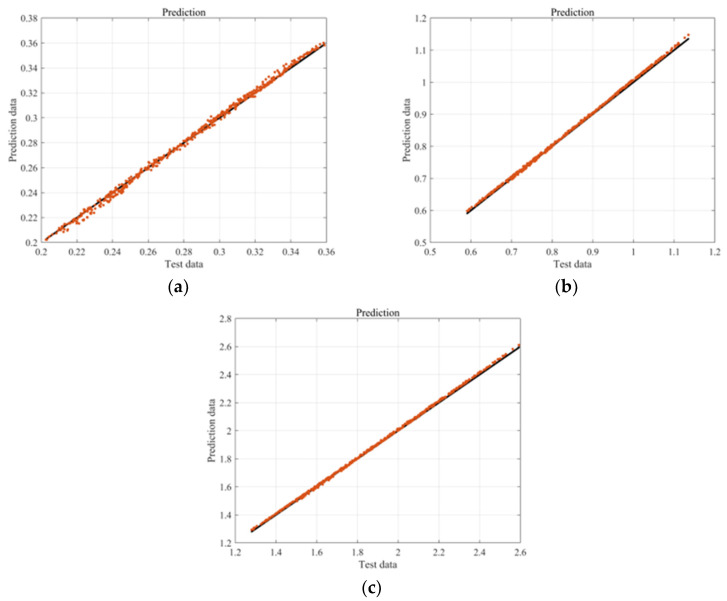
Differences between predicted and actual data (black line: true value, red dot: predict value): (**a**) 1st paired mode; (**b**) 2nd paired mode; (**c**) 3rd paired mode.

**Figure 15 sensors-23-06169-f015:**
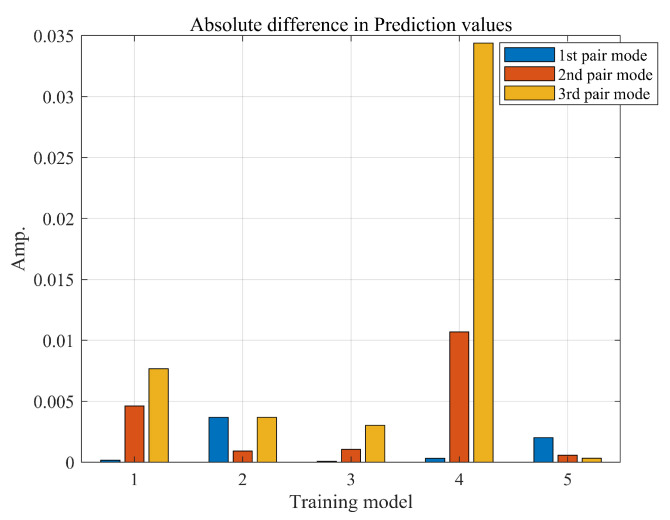
Absolute difference in prediction values.

**Table 1 sensors-23-06169-t001:** Experimental measurements and numerical results for lead screw system.

Simulation [Hz]	Experiment [Hz]	Error [%]	Mode Shape
25.9	26.0	0.4	1st bending pair
28.4	28.0	1.4	
74.2	73.3	1.2	2nd bending pair
79.3	77.3	2.6	
147.0	145.0	1.4	3rd bending pair
156.0	151.0	3.3	

**Table 2 sensors-23-06169-t002:** Nominal value of 3D model.

Parameter	Symbol	Value
Translation distance	d	variable
Rotational speed	ohm	28.64 rpm
Length of lead screw	L	996.301 mm
Young’s modulus	E	240 GPa
Density	ρ	7.96 × 10^9^ ton/mm^3^

**Table 3 sensors-23-06169-t003:** Dataset split ratio.

Data	Percentage	Number of Samples
Training	56%	1400
Validation	14%	350
Testing	30%	750

**Table 4 sensors-23-06169-t004:** DNN model.

Layer	Output Shape	Param#	Connect to
Input Layer	(None, 2)	0	-
Dense 1	(None, 256)	768	Input Layer
Dense 2	(None, 128)	32,896	Dense 1
Dense 3	(None, 64)	8256	Dense 2
Dense 4	(None, 32)	2080	Dense 3
Y1_output	(None, 1)	129	Dense 2
Y2_output	(None, 1)	65	Dense 3
Y3_output	(None, 1)	33	Dense 4

## Data Availability

Data sharing not applicable.

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
