# Peer review of "Analysis of Friction Noise Mechanism in Lead Screw System of Autonomous Vehicle Seats and Dynamic Instability Prediction Based on Deep Neural Network"

_sensors, 2023, doi:10.3390/s23136169_

Round 1

Reviewer 1 Report

I have a few remarks.

The introduction section should focus more on a literature review. Is it possible to write the beginning you wrote (1. Introduction), in which you described the problem under study, in a separate section called “problem formulation”.

Check the text, hyphens appeared in the words.

Can you give specific recommendations on how to reduce the squeal, noise?

Author Response

Reviewer #1

Thank you for your review.

Q1. The introduction section should focus more on a literature review. Is it possible to write the beginning you wrote (1. Introduction), in which you described the problem under study, in a separate section called “problem formulation”.

A1. Problem formulation was separated in the end of introduction section. We have also added a structure of the paper.

 “1.1 problem formulation

In this study, we applied the mode-coupling mechanism [24], which is generally used in the brake system, to the lead screw system of an autonomous vehicle seat and investigated the friction noise that could occur between the lead screw and lead nut. In particular, we aimed to provide a method to understand the vibration/noise issues in the vehicle seat design stage by clearly understanding the squeal mechanism occurring in the lead screw through a friction-coupling integrated 3D model and simplified model. Additionally, we approached friction instability, which had been reliant solely on physical interpretation, from a deep learning perspective, and predicted the dynamic instability through deep learning.

The paper is structured as follows: in Section 2.1 result of single component modal analysis and experiment are verified; in Section 2.2 to 2.4 configuration of FE model, method of complex eigenvalue analysis and results of analysis are described; in Section 2.5 mode-coupling mechanism of lead screw is described; in Section 2.6 Squeal noise is predicted using deep learning; in Section 3, there is a discussion of results, followed by the conclusion.”

Q2. Check the text, hyphens appeared in the words.

A2. hyphens appeared in the words was modified on full sentences

Q3. Can you give specific recommendations on how to reduce the squeal, noise?

A3. This paper describes the mechanism of squeal noise in lead screws. Therefore, we did not focus on optimizing methods to reduce the squeal noise generated by the lead screw. However, the sensitivity map of the friction noise shows that when the modes of the pair are frequency separated due to geometry or boundary conditions, instability due to mode-coupling does not occur.

Reviewer 2 Report

sensors-2466125- Analysis of Friction Noise Mechanism in Lead Screw System of  Autonomous Vehicle Seats and Dynamic Instability Prediction  based on DNN  by Jaehyeon Nam , Soul Kim  and Dongshin Ko

The manuscript deals with the study investigated the friction-induced squeal mechanism in a lead screw system.  A complex eigenvalue analysis was performed through a finite element model and a two-degree-of-freedom model was described to analyze the closed solutions generated in a lead screw and the sensitivity to friction noise was examined. The results of the analysis showed that the main cause of friction noise in a lead screw was bending mode pairs.

-The introduction should be improved, it is not possible to identify the importance and relevance of the topic for the scientific community.

-It is not clear the problem that is being solved, show the context in which the scientific community demands the solution of the problem.

-It is important to clearly define the contribution of the manuscript.

-The experimental results presented in 2.1 are irrelevant to the investigation. The modal analysis performed is not correct, the support  condition does not correspond to the real condition presented by the system under study nor to the geometry and shape of the system.

-L118-126 What is stated in these lines is not clear. What support conditions do you use, why in spite of having a symmetrical bar there are differences in the frequencies? You should check the experimental modal analysis, something is not right. What about the rigid modes?

-The frequencies presented in Figure 7 are not comparable with those found in the experimental analysis.

-It is serious that the manuscript does not have a conclusion section.

-The bibliographic references presented in the manuscript are old and do not represent the state of the art of the subject.

Dear authors, this reviewer feels that the manuscript presents a great deal of work but is not well conducted. It is not really clear how you are directing the work towards achieving the solution to the problem. The manuscript should be completely restructured.

Author Response

Reviewer #2

Thank you for your review.

Q1. The introduction should be improved, it is not possible to identify the importance and relevance of the topic for the scientific community.

A1. Introduction was modified, and reference was added.  

“In this study, we applied the mode-coupling mechanism [24], which is generally used in the brake system, to the lead screw system of an autonomous vehicle seat and investigated the friction noise that could occur between the lead screw and lead nut. In particular, we aimed to provide a method to understand the vibration/noise issues in the vehicle seat design stage by clearly understanding the squeal mechanism occurring in the lead screw through a friction-coupling integrated 3D model and simplified model. Additionally, we approached friction instability, which had been reliant solely on physical interpretation, from a deep learning perspective, and predicted the dynamic instability through deep learning. Therefore, this paper understood the friction noise generated by the lead screw from a mechanism perspective and described various methods to predict the friction noise. In these results, we proposed a way to improve the operating noise of autonomous vehicle seats at the design stage.”

Q2. It is not clear the problem that is being solved, show the context in which the scientific community demands the solution of the problem.

A2. This paper describes the mechanism of the friction-induced vibration generated by the lead screw. The Friction noise has been studied for a long time, but it is one of the cases that has not been clearly solved. In this study, we analyzed the cause of the screech noise generated by the lead screw from the mechanism perspective. We also presented a method for predicting the screech noise using deep neural network.

Q3. It is important to clearly define the contribution of the manuscript.

A3. The introduction has been revised to clarify the contributions of the paper.

Q4. The experimental results presented in 2.1 are irrelevant to the investigation. The modal analysis performed is not correct, the support  condition does not correspond to the real condition presented by the system under study nor to the geometry and shape of the system.

A4. Section 2.1 presents the results of the modal analysis performed on the Free-Free boundary conditions for a single piece. The reason for performing the modal analysis is to calibrate the modulus of elasticity for the analyzed single component model. As shown in Section 2.3, the results of the complex eigenvalue analysis are based on the modal matrix of the system. Therefore, since the results of the mode analysis are very important, we worked to improve the reliability of the analysis by calibrating the analysis model.

Q5. L118-126 What is stated in these lines is not clear. What support conditions do you use, why in spite of having a symmetrical bar there are differences in the frequencies? You should check the experimental modal analysis, something is not right. What about the rigid modes?

A5. Figures 1 and 4 show that the ends of the lead screws are bracketed for system assembly, resulting in an asymmetrical condition for the beam. This leads to a frequency separation between each direction, as depicted in Fig. 2. Under constrained conditions, the frequency difference is smaller compared to the free condition. However, Fig. 5 demonstrates that depending on the nut position, there is a frequency difference between the pair modes of the lead nut. Rigid mode is not considered.

Q6. It is serious that the manuscript does not have a conclusion section.

A6. Discussion and future work have been modified and changed to a conclusion section.

Q7. The bibliographic references presented in the manuscript are old and do not represent the state of the art of the subject.

A7. Recently written articles were added in the references.

“24. Kim, D.; Min, K.; Kim, H.; Huh, K. Vehicle sideslip angle estimation using deep ensemble-based adaptive Kalman filter. Mech. Syst. Signal Process. 2020, 144, 106862.

  1. Tork, N.; Amirkhani, A.; Shokouhi, S.B. An adaptive modified neural lateral-longitudinal control system for path following of autonomous vehicles. Eng. Sci. Technol. Int. J. 2021, 24, 126–137.
  2. Šabanovič, E.; Žuraulis, V.; Prentkovskis, O.; Skrickij, V. Identification of Road-Surface Type Using Deep Neural Networks for Friction Coefficient Estimation. 2020, 20, 612.
  3. Anutcharee, K.; Kiatfa, T.; Supachai, P.; Nuttapong, L. Study of Brake Pad Shim Modification to Improve Stability Against High Frequency Squeal Noise by Finite Element Analysis. Engineering Journal. Sep 2022, 26, 25-34.
  4. Liu, S.; Silva, U.; Chen, D.; Leslie, A.; Meehan, P. Experimental investigation of wheel squeal noise under mode coupling. International Conference on Contact Mechanics and Wear of Rail/Wheel system. Sep 2022, 266-267.

30 Meehan, P.; Leslie, A. On the mechanisms, growth, amplitude and mitigation of brake squeal noise. Mechanical Systems and Signal Processing. May 2021, 135.”

Reviewer 3 Report

This is a very interesting approach to the problem. However, there are three issues that the authors failed to address:

1. The abstract section should be more intensively focused on the main idea directly and must contain the contribution of this manuscript with numerical result indicators.

2. Keywords should include relevant words that capture the essence of your research, they need to be listed in an appropriate order of importance.

3. A more state-of-the-art literature review should be undertaken to cover various applications of the proposed approach.

4. More explication the  single 14 model was constructed using DNN.

5. What is the software used for the simulation part?

6. The conclusion section should be rearranged, and numerical results should be added. Also, the authors may propose some interesting problems as future work in the conclusion.

Author Response

Reviewer #3

Thank you for your review.

This is a very interesting approach to the problem. However, there are three issues that the authors failed to address:

Q1. The abstract section should be more intensively focused on the main idea directly and must contain the contribution of this manuscript with numerical result indicators.

A1. The abstract was modified.

“This study investigated the squeal mechanism induced by friction in a lead screw system. The dynamic instability in the friction noise model of the lead screw was derived through a complex eigenvalue analysis via a finite element model. A two degree of freedom model was described to analyze the closed solutions generated in the lead screw, and the friction noise sensitivity was examined. The analysis showed that the main source of friction noise in the lead screw was the bending mode pair, and friction-induced instability occurred when the ratio of the stiffness of the bending pair modes was 0.9 to 1. We also built an architecture to predict multiple outputs from a single model using deep neural networks and demonstrated that friction-induced instability can be predicted by deep neural networks. In particular, instability with nonlinearity was predicted very accurately by deep neural networks with a maximum absolute difference of about 0.035.”

Q2. Keywords should include relevant words that capture the essence of your research, they need to be listed in an appropriate order of importance.

A2. Keywords was modified.

“Keywords: Friction noise, Mode-coupling mechanism, Squeal sensitivity map, Squeal in-stability estimation, Deep neural network (DNN)”

Q3. A more state-of-the-art literature review should be undertaken to cover various applications of the proposed approach.

A3. state-of-the-art literature was added in the reference.

“24. Kim, D.; Min, K.; Kim, H.; Huh, K. Vehicle sideslip angle estimation using deep ensemble-based adaptive Kalman filter. Mech. Syst. Signal Process. 2020, 144, 106862.

  1. Tork, N.; Amirkhani, A.; Shokouhi, S.B. An adaptive modified neural lateral-longitudinal control system for path following of autonomous vehicles. Eng. Sci. Technol. Int. J. 2021, 24, 126–137.
  2. Šabanovič, E.; Žuraulis, V.; Prentkovskis, O.; Skrickij, V. Identification of Road-Surface Type Using Deep Neural Networks for Friction Coefficient Estimation. 2020, 20, 612.
  3. Anutcharee, K.; Kiatfa, T.; Supachai, P.; Nuttapong, L. Study of Brake Pad Shim Modification to Improve Stability Against High Frequency Squeal Noise by Finite Element Analysis. Engineering Journal. Sep 2022, 26, 25-34.
  4. Liu, S.; Silva, U.; Chen, D.; Leslie, A.; Meehan, P. Experimental investigation of wheel squeal noise under mode coupling. International Conference on Contact Mechanics and Wear of Rail/Wheel system. Sep 2022, 266-267.

30 Meehan, P.; Leslie, A. On the mechanisms, growth, amplitude and mitigation of brake squeal noise. Mechanical Systems and Signal Processing. May 2021, 135.”

Q4. More explication the single 14 model was constructed using DNN.

A4. We used a single model. The architecture, described in Table 4, can predict multiple outputs, namely Y1, Y2, and Y3, within a single model.

Q5. What is the software used for the simulation part?

A5. In the simulation part, we used Abaqus to analyze the squeal instability.

Q6. The conclusion section should be rearranged, and numerical results should be added. Also, the authors may propose some interesting problems as future work in the conclusion.

A6. The conclusion and future work were revised.

“In this study, we constructed a squeal model by applying a vibration instability analysis algorithm, typically used to analyze brake squeal, to a lead screw analysis model, and conducted a vibration instability analysis to examine the squeal mechanism occur-ring in lead screws. We also researched a methodology that could predict not only non-linear results but also multiple outputs through a single model by forecasting the instability of the lead screws owing to geometric shape deformation by using a DNN. According to the results of a complex eigenvalue analysis of the lead screw, the system's instability was manifested in the bending mode pairs. This is because the paired modes of the lead screw characteristically have almost no frequency difference; hence, the mode-coupling sensitivity was large even at a low friction coefficient. The analysis of the mode-coupling sensitivity shows that friction noise generated by the lead screw occurs when the ratio of the two stiffnesses is between 0.9 and 1. The mode-coupling sensitivity is higher when the stiffness ratio is 1. This means that the natural frequencies of the paired modes are completely coincident, suggesting the need to separate the natural frequencies of the paired modes through constraints and geometry changes in order to stabilize the friction noise. We also demonstrated that the single architecture multi output model using DNN could rapidly predict the instability of models with extreme nonlinearity, such as geometric shape nonlinearity. The analysis showed that the absolute difference between the predicted values was very accurate, up to 0.035. This implies that even a simple DNN model can be used to conduct complex instability analyses of systems containing nonlinearity.

 In future work, we intend to conduct studies applying machine learning to transient analysis that can predict the actual behavior of friction noise. There are two main ways to estimate friction-induced vibration. The first method can be predicted to the onset time of dynamic instability using the linearization. On the other hand, the second method are calculated by differential equation to consider by all of its non-linear characteristics. A direct solution to the differential equation can yield a more accurate solution, but the non-linearity makes the computation complex and time consuming. Therefore, we plan to use machine learning to approximate the solution of the friction-induced vibration.”

Reviewer 4 Report

The paper investigated a squeal model by applying an analysis algorithm of vibration instability, used to analyze brake squeal, to a lead screw analysis model, and conducted a vibration instability analysis to examine the squeal mechanism occurring in lead screws. The authors also worked out a methodology forecasting the instability of the lead screws owing to geometric shape deformation by using deep neural network method that could predict not only nonlinear results but also multiple outputs through a model.
The publication contains generally new scientific results.
Number of referred publications is 30, which is enough and advisable.
The following corrections would improve the quality of paper:
-  Please, depict structure of paper at the end of section Introduction (e.g.: “The rest of the paper is organized as follows: …”).
-  Please, avoid acronyms in the titles of the paper and sections.
-  There are lots of acronyms, so I suggest doing a Nomenclature section. For example: What is the DNN? - Not all readers are experts in this specific field!
- There are lots of typos:
For example: “for-ward” row 25; “mod-el” row 83; “com-plex” row 147, etc.
I suggest using same size letters (maybe same as text)  in figures (see Figure 8, left: too big letters; right: too small letters).
For non-dimensionalization, the authors introduced a “Tau” variable in line 274, which is not used later. What is the reason for its introduction?
Section 3 should be expanded by deductions in more details and suggestions for future works.

Author Response

Reviewer #4

Thank you for your review.

The paper investigated a squeal model by applying an analysis algorithm of vibration instability, used to analyze brake squeal, to a lead screw analysis model, and conducted a vibration instability analysis to examine the squeal mechanism occurring in lead screws. The authors also worked out a methodology forecasting the instability of the lead screws owing to geometric shape deformation by using deep neural network method that could predict not only nonlinear results but also multiple outputs through a model.

The publication contains generally new scientific results.

Number of referred publications is 30, which is enough and advisable.

The following corrections would improve the quality of paper:

Q1. Please, depict structure of paper at the end of section Introduction (e.g.: “The rest of the paper is organized as follows: …”).

A1. The structure of paper was added in the end of section Introduction

“The paper is structured as follows: in Section 2.1 result of single component modal analysis and experiment are verified; in Section 2.2 to 2.4 configuration of finite element (FE) model, method of complex eigenvalue analysis and results of analysis are described; in Section 2.5 mode-coupling mechanism of lead screw is described; in Section 2.6 Squeal noise is predicted using deep learning; in Section 3, there is a discussion of results, followed by the conclusion.”

Q2. Please, avoid acronyms in the titles of the paper and sections.

A2. Acronyms was modified to nomenclature in title and paper section.

“Title: Analysis of Friction Noise Mechanism in Lead Screw System of Autonomous Vehicle Seats and Dynamic Instability Prediction based on Deep Neural Network”

“2.2. Finite Element Model”

“2.3. Method for complex Eigenvalue Analysis of Finite Element Model”

“2.6 Deep Neural Network based Squeal Instability Estimation”

Q3. There are lots of typos:

A3. We've found and fixed typos throughout the documentation and corrected the text in the figures.

Q4. For non-dimensionalization, the authors introduced a “Tau” variable in line 274, which is not used later. What is the reason for its introduction?  

A4. Non-dimensionalization was changed to dimensionless time. The equations of motion (10~) are written using dimensionless time to eliminate unnecessary variables and define meaningful variables. Dimensionless also makes it easier to describe changes in generalized parameters.

Q5. Section 3 should be expanded by deductions in more details and suggestions for future works.

A5. More details about future work were added in Section 3.

“In future work, we intend to conduct studies applying machine learning to transient analysis that can predict the actual behavior of friction noise. There are two main ways to estimate friction-induced vibration. The first method can be predicted to the onset time of dynamic instability using the linearization. On the other hand, the second method are calculated by differential equation to consider by all of its non-linear characteristics. A direct solution to the differential equation can yield a more accurate solution, but the nonlinearity makes the computation complex and time consuming. Therefore, we plan to use machine learning to approximate the solution of the friction-induced vibration.”

Round 2

Reviewer 2 Report

sensors-2466125- Analysis of Friction Noise Mechanism in Lead Screw System of  Autonomous Vehicle Seats and Dynamic Instability Prediction  based on DNN  by Jaehyeon Nam , Soul Kim  and Dongshin Ko

Dear authors, in spite of the improvements made to the manuscript, I consider that the work developed is not well directed towards the solution of the problem. The amount of work presented is recognized but it does not have a systematic logic towards the solution of the problem that you propose.  It is not really clear how you are directing the work towards achieving the solution to the problem. 

Author Response

Reviewer #2

Thank you for your review.

Q1. Dear authors, in spite of the improvements made to the manuscript, I consider that the work developed is not well directed towards the solution of the problem. The amount of work presented is recognized but it does not have a systematic logic towards the solution of the problem that you propose.  It is not really clear how you are directing the work towards achieving the solution to the problem. 

A1. The introduction has been revised to provide a clear direction towards solving the problem addressed in the paper and to highlight the aim of the study.

“In this study, we applied the mode-coupling mechanism [27-30], which is generally used in the brake system, to the lead screw system of an autonomous vehicle seat and investigated the friction noise that could occur between the lead screw and lead nut. In particular, we aimed to provide a method to understand the vibration/noise issues in the vehicle seat design stage by clearly understanding the squeal mechanism occurring in the lead screw through a friction-coupling integrated 3D model and simplified model. The analytical model was validated by the results of the modal analysis of the lead screw in free boundary condition. Then, the friction noise prediction was performed by complex eigenvalue analysis of the lead screw system. To analyze the mechanism of friction-induced vibration in a leadscrew system, a two-degree-of-freedom model was constructed. Then, the friction noise in the lead screw system was analyzed by the mode-coupling mechanism. Additionally, we approached friction instability, which had been reliant solely on physical interpretation, from a deep learning perspective, and predicted the dynamic instability through deep learning. Therefore, this paper understood the friction noise generated by the lead screw from a mechanism perspective and described various methods to predict the friction noise. In these results, we proposed a way to improve the operating noise of autonomous vehicle seats at the design stage.”
